# Energy aware stable path ad hoc on-demand distance vector algorithm for extending network lifetime of mobile ad hoc networks

**Tibebu Legesse[1], Dagne Walle Girmaw ID [2]\*, Esubalew Yitayal[3], Engida Admassu[4]**

**1** Department of Information Technology, Wollo University, Dessie, Ethiopia, **2** Department of Information Technology, Haramaya University, Harar, Ethiopia, **3** Department of Electrical and Computer Engineering, Debre Birhan University, Debre, Ethiopia, **4** Department of Information Systems, Mizan-Tepi University, Mizan-Aman, Ethiopia

\* dagnewalle143@gmail.com

## Abstract

Traditional routing protocols in MANETs, such as AODV, do not take energy consumption into account, which leads to inefficient utilization of the limited battery resources. In such networks, nodes depend on non-rechargeable batteries, and once the energy is depleted, the nodes can no longer function, resulting in frequent link failures and reduced network lifetime. To address these challenges, this paper proposed a novel routing protocol, the Energy Aware Stable Path Ad Hoc On-Demand Distance Vector (EASP-AODV), which incorporates residual energy of individual nodes and received signal strength to improve routing decisions. By considering these factors, EASP-AODV effectively reduces link breakages, enhances route stability, and prolongs the overall lifetime of the network. The algorithm optimizes energy usage, ensuring that nodes with higher residual energy are preferred for routing, thus minimizing energy depletion and mitigating early node failures. To evaluate the effectiveness of EASP-AODV, we conducted simulations using NS2.35, a widely used network simulator. The performance of EASP-AODV was compared with two well-known routing protocols: Energy Aware Path Selection (EAPS-AODV) and the traditional AODV. Several performance metrics, including normalized energy consumption, packet delivery ratio, normalized routing overhead, average end-to-end delay, and network lifetime, were measured. Simulation results demonstrate that EASP-AODV outperforms EAPS-AODV and AODV in terms of energy efficiency, with a 4.5% and 9.5% reduction in normalized energy consumption, respectively. Moreover, the proposed algorithm significantly extends network lifetime, with a **3.5%** and **7.5%** improvement over EAPS-AODV and AODV. These results confirm that EASP-AODV provides a more energy-efficient and sustainable solution for routing in MANETs, addressing the critical issues of energy consumption and network stability while extending the operational lifetime of the network.

## 1. Introduction

A network is a collection of two or more devices that are linked together to share information. In practice, it is a collection of different computer systems linked together via a physical or

**Data availability statement:** All relevant data are within the article.

**Funding:** The author(s) received no specific funding for this work.

**Competing interests:** The authors declare that they have no competing interests.

wireless connection. Numerous technological advances have been made in the field of wireless communication networks in recent years [1]. Fig 1 depicts infrastructure-based and infrastructure-less networks in this type of communication network.

Mobile Ad-hoc Networks (MANETs) are infrastructure-less networks that are formed on an ad-hoc basis, self-configured, and self-controlled. They have dynamic behavior and are formed without centralized administration. Communications between participating nodes take place hop by hop. MANET's role is to provide internet connectivity to nodes anywhere and at any time, regardless of geographical location. MANETs are multi-hop networks with a dynamic topology. According to an open standard organization, called Internet Engineering Task Force (IETF), the working group defines MANET as an Autonomous system of mobile routers (associated hosts) connected by wireless links as shown in Fig 2.

Because the routers are free to move randomly and organize themselves arbitrarily, their topology changes quickly and unexpectedly. A network of this type can operate independently or be linked to the larger Internet [2].

The node acts as a router by itself, forwarding and receiving packets. Link failure occurs due to the dynamic nature of the system and the lack of a central controlling mechanism. Each node serves as both a host and a router. So, routing is the process of selecting paths for data sent from a source to a destination. Given the limitations of MANETs, the path selection processes are traditional communication methods that select the shortest path.

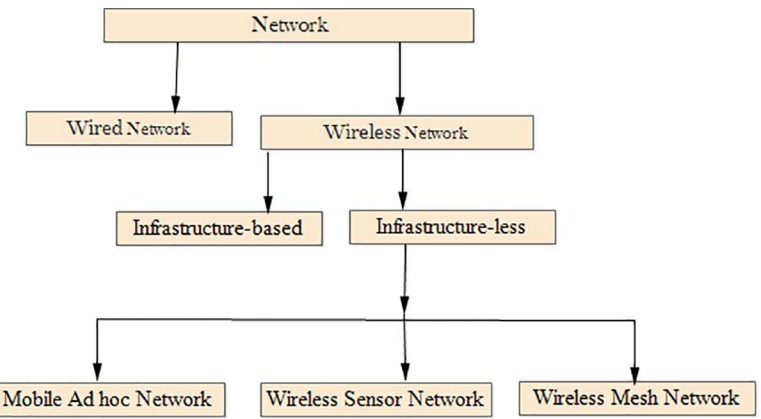

**Fig 1. Classification of computer networks.**

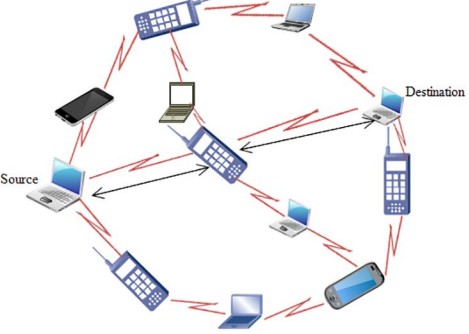

**Fig 2. General mobile ad hoc network topology.**

However, an optimized route and stable path selection are required. MANET constraints include limited battery power, low bandwidth, and high error rates. The most difficult issue for link breakage is a lack of node battery power. Sending and receiving packets consumes more energy during the routing process. The nodes have no way of getting battery energy to perform their functions; they can only get power from the attached battery. If a node lacks sufficient energy, packet transmission and reception are halted, resulting in link failure [3]. Routing is one of the energy-consuming processes in a MANET. Nodes can waste energy during packet flooding, and nodes with limited battery life cannot transmit data for long periods [4]. Packet loss is being caused by link failure. Because the nodes are energy-operating nodes, they lack a mechanism for obtaining batteries, implying that there is no replacement technique. Simply put, the node can draw power from the attached battery. As a result, node energy consideration is a critical activity for maximizing energy utilization and extending network lifetime in MANETs.

The **motivation** behind the proposed routing method lies in the **energy efficiency** challenges inherent in MANETs. Traditional routing methods focus on selecting the shortest path but do not account for the dynamic energy consumption and link stability issues that arise in such environments. As nodes in a MANET lack a mechanism for recharging their batteries, energy depletion can lead to **link breakage**, **packet loss**, and eventual network partitioning. Therefore, it becomes crucial to develop routing strategies that not only minimize the number of hops but also prioritize **energy conservation** to enhance **network lifetime**.

Mobile Ad-hoc Networks (MANETs) have gained significant interest due to the expansion of wireless devices and networks. As infrastructure-less wireless communication networks, MANETs offer dynamic mobility of nodes without geographical restrictions, extending services in infrastructure-based networks. However, challenges such as dynamic topology changes, energy constraints, bandwidth limitations, QoS issues, high end-to-end delay, and routing overhead persist. Among these, energy constraints remain an open area of research that is not fully addressed in existing MANET routing protocols. Efficient and stable paths are required for long transmission and continuous data forwarding. This research focuses on reducing link breakage caused by limited node energy and extending the network lifetime of MANETs.

Authors in [5], presented a signal strength and residual power-based optimum transmission power routing approach for MANETs. In this method, path selection is based on low residual power parameters and incoming packet power strength with variable power transmission techniques and mobility of node distance (far and nearest) consideration, so different nodes participate in RREQ broadcast. This leads to increased control overhead and node energy consumption. According to the researchers, they focused on node distance but ignored individual node energy and overhead, which occurs when a link breaks. Researchers in [6] proposed an energy-aware MANET based on efficient AODV. The average residual energy of the path is used to select the best path in this algorithm. The algorithm performs the routing process by considering nodes with a middle value of received signal strength. The nodes in the middle zone are responsible for forwarding the RREQ to the destination. The destination node can choose the path with the highest average energy. At the time, the individual node battery was not taken into account. The path only considers the total path energy and chooses the path with the highest average energy. As a result, the chosen path will not have a longer network lifetime, which means that the node may exhaust its energy before sending the actual data, resulting in link breakage. Individual node residual energy and received signal strength were taken into account in our paper. As a result, our proposed approach extends the network lifetime.

The rest of the sections are organized as follows: Section 2 presents the related works and in section 3, we present the proposed algorithm. Section 4 presents the results and discussion, and finally, the conclusion and future work are presented in Section 5.

## 2. Related works

Authors in [3] presented a residual battery capacity-based routing protocol for extending the lifetime of MANETs. In this work, the authors describe the dynamic nature of MANET and how routing of packets is a challenging task. When the network is divided or partitioned in the case of mobility, the node battery leads to link breakage. As a result, the authors concluded that an energy-aware routing algorithm is required for MANET. The authors' main focus here was to simulate existing MANET routing protocols based on various metrics such as energy consumption, packet delivery ratio (PDR), throughput, overhead, end-to-end delay, and network lifetime. The path was chosen by the destination node by comparing the residual energy of all other nodes that are paired with the source and destination. The results show that the dynamic source routing (DSR) algorithm outperforms the AODV algorithm in small networks, but in large networks, network lifetime remains an issue. Authors in [4] proposed a location-based energy-efficient scheme for maximizing the routing capability of AODV protocol in MANET that uses energy and reduces packet loss due to link breakage. This study reduces destination finding by employing the location-aware concept and recording each node's residual energy, location, and speed. In the study, nodes with battery energy greater than the threshold value will forward packets using distance combination techniques, and the results show that packet losses were reduced. When only nodes with energy greater than the threshold value are taken into consideration, it leads to the possibility of overhead because if the number of nodes participating in the route is large, all of them will flood, resulting in overhead, which leads to delay in recording the location of other nodes. The authors in [5] proposed an optimum transmission power routing approach for MANETs based on signal strength and residual power. The RSS is computed using the Friss transmission equation and the remaining residual energy and received signal strength were used in this work. While receiving a packet, the transmitter and receiver calculated the received signal strength to estimate the node's location. The results show that the network lifetime is increased while the energy consumption rate of mobile nodes is reduced with the least amount of end-to-end delay. Even if it has a solution to reduce link breakage, there is still overhead due to all node participation. In [6], the authors presented an energy-aware path Selection Efficient AODV for MANET. In this proposed method, the network around the source node range is divided into three zones to reduce control overhead, but the destination node selects the path based on average energy, which causes link breakage due to individual node energy inconsideration, so a stable path selection algorithm is required for creating stable communication. In [7], the authors proposed an energy-efficient path selection algorithm in MANET, in which they discussed critical routing protocol issues and designed new modified algorithms based on the residual energy of the node. The changes are made during the node's route discovery time, which is when the source node wishes to communicate with the destination, as well as before processing the RREQ by using the node's received signal strength and residual energy, after processing the RREQ, and after broadcasting the RREQ in terms of the jitter imposed on the RREQ message. The results show that combining modified DSR and AODV, namely MDSR and MAODV, reduced MANET link breakage. However, because it takes total energy into account, this method has no long path establishment method. The authors in [8] proposed an AODV-based energy-efficient routing protocol for maximizing MANET network lifetime. The proposed algorithm demonstrates that the lack of central administration is a

source of energy consumption in MANET. The proposed work is based on the network's mean energy. The work has two main considerations: (a) the number of paths between the source and the destination, and (b) the number of nodes participating in each path. Based on these considerations, the destination node can calculate the path's mean energy and then unicast the RREQ message along the reverse path of the first RREQ message received. The simulation results show that the network lifetime is significantly longer than the existing protocol when using NS2. However, the proposed work takes into account the network's energy mean value, which includes the total number of nodes and the path within the network that leads to packet loss. An energy-aware simple ant routing algorithm for MANET was proposed in [9]. In the work, ant colony base routing was used. Ant colony routing (ACR) is a distributed routing technique that is based on swarm intelligence, where the foraging behaviors of ants are used in solving the optimization problem. The authors named the presented algorithm a Simple routing protocol (SARA). It employs two methods: forward Ant (FANT) and backward Ant (BANT), the primary goal of which is route establishment. The residual energy of the node is incorporated for the cost function of SARA during route discovery time. According to the simulation results, the network lifetime can be extended by selecting the nodes with the highest residual energy during the route discovery process. The researchers used packet delivery ratio, energy consumption, thought, end-to-end delay, and normalized routing overhead as evaluation metrics to assess the performance of their algorithm. The main disadvantage of the paper is that it does not take individual nodes into account and does not use normalized energy consumption and network lifetime as performance evaluation metrics. The authors in [10] presented a MANET-based mobile agent-based energy efficient reliable routing protocol. The proposed approach employs a link cost metric for the first time, as well as network load in terms of node burthen degree and bandwidth usable degree, minimum drain rate for energy consumption, and link availability.

The mobile agents are organized at random and transfer data hop by hop until they arrive at their destination node. The above-mentioned metrics and a combined list of metrics collect information for each hop they traverse based on this hop-by-hop communication. The main goal of this algorithm is to find the best path by using the path cost metric, which is the sum of the link cost metric along the path. The authors used the average packet delivery ratio, average end-to-end delay, throughput, and packet drop to evaluate the proposed algorithm. In general, multiple paths are established after collecting information from agents, and the source node selects the optimal path using the path cost metric, but no energy performance evaluation metrics such as normalized energy consumption were used. The authors in [11] presented a new energy-based power-aware routing method for MANETs. The main problem to be solved was link failure, which caused packet loss. To address these issues, the proposed algorithm employs the shortest distance path method with the highest residual energy of the node. To evaluate this algorithm, the authors used quantitative performance metrics such as network lifetime, throughput, packet delivery ratio, packet loss ratio, and end-to-end delay. As a result, the proposed dynamic energy ad hoc on-demand distance vector (DE-AODV) protocol outperformed other techniques that are currently available. This paper employs several energy preservation techniques, but this technique not only increases the energy level but also maximizes the MANET network lifetime. The main disadvantage, according to the paper, was that only the highest node residual energy was considered and they did not consider energy consumption performance metrics. The authors in [12] presented a novel energy-efficient obstacle-aware routing algorithm for MANETs. In the paper, the authors discussed MANET route failures caused by the presence of obstacles. Stable route searching consumes a lot of energy; as a result, many nodes fall below the threshold value, causing the node to become inactive. The performance of a mobile ad hoc network suffers as a result of such a problem. As

a result, the authors proposed a novel energy-efficient and obstacle-aware routing algorithm (EEOARA) for addressing obstacles in real-world MANET operations to address this issue. The authors used maximum total residual energy; this is accomplished by appending node energy and calculating the average total path energy; the maximum total path energy is then chosen from among the paths. Routing overhead, packet delivery ratio, average end-to-end delay, and average energy consumption are the primary algorithm performance metrics. The main drawback of the energy efficient aware method is individual node energy in consideration because the maximum total energy of the path is computed by the sum of the smallest residual energy and highest residual energy of the node, so this technique still has link failure, which means low energy nodes are involved in the path selection and also the packet delivery ratio is low, indicating link failure. The authors in [13] presented a MANET stability and energy-aware reverse AODV routing protocol. The primary goal of this research paper is to reduce routing overhead. This was accomplished through the use of two methods: energy and reliability factor (RF). In this method, the route is chosen based on the highest reliability factor (RF). During route selection, the reliability factor for each path is calculated using Route Reply Latency (RRL) time. When the destination node receives more than one RRREQ packet from the same next hop address before the timer t1 expires, the routing table is updated. During packet forwarding, the number of route entries made by the source node for a specific destination node equals the number of next hop nodes. So forward routing is done with the highest RF value, which means the highest reliable factor value. If the primary path fails, different secondary paths are used in descending order of their RF values. The packet delivery ratio is reduced as a result of the simulation, but they did not use normalized energy consumption as a performance metric, because this metric tells how much energy is consumed during packet transmission. EDA-AODV: Energy and Distance Aware AODV Routing Protocol was proposed by the authors of [14]. Because EDA-AODV is a new version of a standard AODV routing algorithm, the presented algorithm primarily focused on modifying AODV route discovery, to obtain stable route selection while using the least amount of node energy, as well as minimizing hop count and flooding of RREQ packets. So, there are two techniques discussed in this paper: node energy and node distance from the transmitting node to the receiving node. The disadvantage of this approach is that when the nodes append their energy during route selection, they do not compare the individual node energy, so all paths are sent as is, which increases destination node overhead and may cause more delay occurrences.

Route Stability and Energy Aware based AODV in MANET was proposed in [15]. Firstly, the Authors mentioned the main challenges that cause stable path selection between source and destination nodes in MANET are depletion of energy and node mobility. As a result, RSEA-AODV was proposed based on the received signal strength of the packet, delay during route discovery, node remaining energy, and node draining rate. The route discovery process involves adding nodes to the path based on received signal strength with a predetermined threshold value. If the nodes are closer to each other, the received signal strength is greater than the given threshold value, and the node can participate in the route discovery process; otherwise, the packet is discarded. The authors also used other parameters such as node draining rate, delay during route discovery, and remaining energy of the nodes. The goal of this work is to extend the network lifetime of the MANET. The work is good for path selection in this case, but the method does not take into account individual node remaining energy.

The authors in [16], proposed centralized genetic-based clustering protocol that faces significant challenges when applied to MANETs, including its reliance on centralized control, high communication overhead, and limited adaptability to node mobility and energy.

The congestion-aware routing and fuzzy-based rate control mechanism study demonstrates effectiveness in WSNs, but its direct applicability to MANETs **is** limited due to

assumptions of static topologies and the computational overhead required for real-time operation [17].

The cluster-based routing method proposed uses mobile sinks to optimize data collection in WSNs, but its applicability to MANETs is limited by assumptions about predictable mobility, scalability challenges, and high energy consumption [18].

The two-level clustering approach using fuzzy logic and content-based routing proposed work shown in IoT but faces challenges in MANETs due to its static assumptions, scalability limitations, and inability to handle the dynamic nature of node mobility. It struggles to handle the dynamic nature of node mobility in MANETs, where nodes frequently change their positions, causing frequent re-clustering and instability in the routing paths. This results in increased overhead and energy consumption. Lastly, the fuzzy logic mechanism, although useful in IoT, may not adapt quickly enough to the rapid changes in a MANET's topology [19].

The gray system theory-based routing protocol for energy consumption management proposed works well for IoT networks, but faces significant challenges in MANETs, including the dependence on centralized cloud infrastructure, dynamic topology, and high computational overhead [20].

The overlapping routing approach **is** proposed for IoT ecosystems faces challenges when applied to MANETs, including its reliance on centralized fog and cloud infrastructure, scalability issues, and potential for increased energy consumption and latency. This approach can lead to increased energy consumption due to the need for continuous communication with central servers [21].

The researcher in [22], proposed a novel approach to message delivery in urban Vehicular Ad Hoc Networks (VANETs), which face challenges due to the highly dynamic mobility of vehicles and limited network infrastructure. Traditional routing protocols often struggle with high latency and low delivery reliability in these environments, especially in cities with unpredictable traffic patterns. To address this issue, the authors introduce a bus-trajectory-based, street-centric routing scheme, leveraging the predictable movement of public transportation buses to improve message delivery efficiency.

Their approach utilizes pre-recorded bus trajectories to model the movement patterns of buses along fixed routes, thus enhancing the reliability of routing decisions. The proposed routing algorithm focuses on a street-centric model, which emphasizes the road network and bus trajectories instead of relying solely on vehicle-to-vehicle communication. This innovation reduces the complexity of routing decisions and improves overall message delivery performance. Through simulations, the authors demonstrate that their method significantly outperforms traditional VANET routing protocols, showing improvements in both message delivery ratio and delay.

Moreover, the proposed bus-trajectory-based routing scheme is scalable, making it suitable for large urban environments with dense traffic. It adapts to varying traffic conditions and limited infrastructure, offering a robust solution for urban message delivery. The paper highlights the potential of this approach for enhancing communication in smart cities, where efficient and reliable message delivery is essential. In conclusion, the research offers a promising solution to VANET routing in urban environments, showing that bus-trajectory-based models can provide more reliable and efficient communication compared to conventional methods.

In [23], proposed a new task scheduling approach in Space-Air-Ground Integrated Networks (SAGINs), aiming to optimize resource allocation for diverse communication tasks. SAGINs, which combine satellite, aerial, and terrestrial networks, are seen as key enablers for next-generation communication systems, but effective task scheduling in such heterogeneous environments remains a challenging problem. The authors propose a proportional

fairness-aware task scheduling mechanism, which ensures that tasks are assigned to resources in a way that balances both system efficiency and fairness among users.

The key innovation of the paper lies in its design of a fairness-aware scheduling algorithm that adapts to the varying network conditions and resource availability across space, air, and ground segments. By incorporating proportional fairness principles, the approach seeks to allocate resources in a manner that maximizes the overall system performance while ensuring that no task or user is unfairly penalized. The authors evaluate their scheme through simulations, demonstrating that their method significantly improves throughput and fairness compared to traditional scheduling techniques.

Additionally, the paper explores the trade-offs between fairness and efficiency, providing insights into how these factors can be dynamically adjusted based on the system's operational conditions. The proposed scheduling scheme is not only applicable to SAGINs but can also be extended to other multi-tier networks with diverse resource constraints. In conclusion, this work advances the state-of-the-art in task scheduling for integrated networks, highlighting its potential for improving communication reliability and fairness in complex, multi-layered environments.

The authors in [24], introduces a new design framework called Lasagna for air-ground integrated infrastructures, aimed at improving the efficiency and flexibility of communication networks in next-generation systems. The authors recognize the growing need for hybrid networks that combine aerial and terrestrial communication elements, which is crucial for applications such as smart cities, emergency response, and IoT. Lasagna is designed to optimize resource allocation, enhance connectivity, and provide seamless integration between air and ground components of the network.

A key feature of the Lasagna architecture is its layered approach, where aerial and terrestrial resources are organized in distinct layers that interact efficiently, allowing for dynamic and adaptive network management. The paper explores why this model works effectively by focusing on its ability to balance network load, reduce latency, and improve coverage. Lasagna aims to overcome the limitations of traditional architectures by providing scalable solutions for communication between drones, ground stations, and mobile devices.

Through simulations and analytical models, the authors demonstrate the system's superior performance in terms of throughput, reliability, and flexibility compared to existing air-ground integration solutions. The paper also highlights how Lasagna can support a wide range of applications, from real-time data streaming to critical communication in disaster recovery scenarios. In conclusion, the Lasagna design represents a promising approach to next-generation air-ground integrated infrastructures, providing both theoretical insights and practical solutions to improve network performance in complex environments.

Authors in [25] presented an AODV RR maximum transmission range based ad hoc on-demand distance vector routing method for MANETs. The work centered on reducing the network's overall energy consumption and communication overhead. Based on their findings, it is preferable to extend the network lifetime of MANETs. This proposed algorithm uses low transmission power, and the routing strategy must be controlled, as well as only certain nodes being allowed or restricted in route request processing, which is accomplished by using received signal strength. As a result, this technique aids in reducing overall network overhead and energy consumption. The authors used packet delivery ratio, routing load, end-to-end delay, throughput, and average consumed energy to evaluate the presented algorithm. This work demonstrates an improvement in network overhead reduction. However, they do not use normalized routing overhead and do not divide the network into inner, middle, and outer zones, which is better for reducing network overhead. Authors in [26] proposed an average link stability with energy-aware routing protocol for MANETs algorithm;

the main idea of this paper is link breakage reduction. The authors presented two techniques in this article: residual energy and link lifetime. The algorithm checks threshold values of residual energy and link lifetime before RREQ is transmitted between sending and receiving nodes. As a result, if the residual energy of the node and the link lifetime are greater than the threshold values, the route request is reprocessed; otherwise, the RREQ is discarded, extending the network lifetime of MANETs. The authors used end-to-end delay, packet delivery ratio, network lifetime, total data sent, total received data average throughput, and total data drop to evaluate the proposed algorithm. According to the simulation result, link breakage is reduced and the network lifetime is extended. However, it does not use energy consumption performance metrics. The authors in [27] presented a routing protocol for MANET, which helps to reduce a node's transmission power for active and inactive routes if the next node is closer. The main goal of this paper is to reduce energy consumption by lowering a node's transmission power. The authors calculated the distance between two consecutive nodes using RSS (received signal strength); at this time, the threshold value is $-75$dBm. The AODV protocol has been modified in three stages as a result of their findings. The first step is to establish a route, which entails forwarding route requests and comparing the current RSS threshold value to the received signal value. The comparison determines whether the node can function as a forwarding node. The second phase is route handling, which involves making changes to the route reply processing and forwarding processes. As a result, the signal's current RSS value is compared to the threshold value. It states that the transmission power of nearby nodes is reduced during the route reply phase. The third phase is route termination, which involves making changes to the route expiry process, which resets the transmission power of the node. In general, if the RSS exceeds a certain threshold during the route request phase, that node will consider forwarding the packet. If the RSS is high during the response phase, it indicates that the nodes are closer together and therefore require less transmission power to send data. As a result of including that modification in the AODV protocol, they reduced the transmission power of the node, which reduces battery consumption, interference, and network battery lifetime. The evaluation metrics for performance are average end-to-end delay, throughput, average jitter, and residual battery. Based on the simulation results, their work demonstrated that the algorithm reduces battery consumption and increases network lifetime, but does not consider nodes with less energy because the authors only consider transmission power and do not use network lifetime and normalized energy consumption as evaluation metrics.

To address these issues, we proposed a maximum of minimum techniques with a received signal strength energy-aware algorithm for improving the MANET network lifetime.

## 3. Proposed methodology

In the EASP-AODV algorithm, individual node residual energy is monitored by tracking energy consumption during communication activities like transmission and reception. Each node updates its energy after each operation, and the residual energy is included in routing requests (RREQ). Nodes with higher residual energy are preferred for route selection to ensure efficient data transmission. This approach helps prevent early node failures due to low energy. In this work, we have considered individual node residual energy and received signal strength. The following assumptions were made:

**Assumptions:**

- The network is densely populated.

- Each node appends its residual energy: - first, before appending its energy, each node compares its current residual energy with incoming node residual energies. If the current node

residual energy is less than the incoming node residual energy, then it appends the current node residual energy, but it does not append the incoming node residual energy.

- The destination node searches for the minimum energy of the path.

- The proposed algorithm uses upper and lower threshold values. We assumed that the RREQ is broadcasted and the route request is received by the intermediate node. In this case, before the intermediate node rebroadcasts it, the incoming RREQ packet signal transmission power is checked whether it's between the lower threshold value of $-78$ and the upper threshold value of $-27$ [6]. The negative ($-78$), indicates the transmission signal is going to be low and $-27$ indicates that the RREQ transmission power is high which means the node is the nearest, so this threshold value only limits the number of middle zone nodes that are participating in the RREQ processing.

**Steps:**

- The source node broadcasts the RREQ.

- The intermediate node receives the RREQ and checks the received signal strength based on the threshold value, and also checks the route to the destination, if known, then sends RREP to the source. If there is no path, it appends its gy by comparing the incoming node energy and sends the RREQ towards the next node.

- Now, the destination node selects the maximum of minimum residual energy (Max (min-energy)) path.

- Then, finally, the destination node sends the RREP towards the source with the highest path energy. The **ECCAD model** optimizes routing by prioritizing nodes with higher residual energy, ensuring balanced energy consumption and reducing link failures. It adjusts routing decisions dynamically based on individual node energy levels, enhancing network stability. This approach prevents early depletion of critical nodes, extending the network's lifetime. By considering energy, the model ensures efficient resource utilization. Ultimately, it maximizes network performance and sustainability

### 3.1. EASP-AODV algorithm for route discovery at the intermediate node

The lines ofpseudo-code presented in Table 1 describe when the source node broadcasts the RREQ to get the best path to reach the destination node. Table 1 presents the EASP-AODV algorithm for route discovery at the intermediate node. The proposed algorithm is shown in Fig 3.

MANETs have a dynamic topology and no fixed node deployment, which may contribute to link failure or breakage. The nodes are isolated and free to join the network. During communication, packet loss occurs with limited node energy. Most of the time, link breakage occurs because of the lack of individual energy consideration which means a a lack of consideration of individual energy levels when selecting the path. Therefore, the proposed algorithm considers individual node energies for stable path selection at the destination node. This is depicted in Fig 4.

### 3.2. AODV algorithm for route selection at node destination

As thepseudo-code section 3.1, which shows how the RREQ is received by the destination node and before RREP is prepared to be sent back to the source node via the reverse route path, the following algorithm further explains the process. The path selection is done with minimum node energy consideration. The AODV algorithm for route selection at node destination is presented in Table 2.

**Table 1. EASP-AODV algorithm for route discovery at the intermediate node.**

| EASP-AODV Algorithm for Route Discovery at the Intermediate Node |
| --- |
| **Begin** |
| **For** all nodes in the network **do** |
| Initiate RREQ to send the data |
| Append their energy while forwarding the RREQ |
| Except for SN and DN, each node calculates the received signal strength and checks the threshold value |
| **End for** |
| **SN** broadcast the RREQ, then |
| IN receives RREQ()//The Intermediate node received the RREQ |
| **If** the IN has a route to DN, then |
| Send RREP()// Then send RREP back to the source node,**Else** |
| The IN checks the RSS and appends their RE |
| If (RSS <= -27 $$ RSS>= -78) then, MZN check their min-energy |
| If (CNE <= INE \|\| INE <= CNE) |
| append (CNE)// append the current node residual energy, **Else** |
| append (INE)// append the incoming node residual energy, then |
| forward RREQ()// flooding RREQ, which is between UTV = -27 and LTV = -78 |
| Else |
| Discard RREQ()// If the RSS is above -27 which is IZN and below -78 which is OZN |
| End If |
| End If |

The intermediate node receives the control message after the source node broadcasts the RREQ. Before rebroadcasting the incoming message, the intermediate node calculates or checks the received signal strength to see if it is the lower, middle, or upper should value. Then, if the nodes are in the middle zone, they append their energy to the control packet and rebroadcast it to the destination.

**3.2.1. Mathematical model for minimum and maximum Path Energy Selection.** Firstly, the node residual energy is computed by comparing the current node residual energy and the incoming node residual energy along the path, to determine the minimum residual energy. This is represented as follows:

$$n1, \; n2, \; n3, \ldots, \; nk \tag{1}$$

Where n denotes the number of nodes in the path.

$$ren1, ren2, ren3, \ldots, renk \tag{2}$$

Where re is the **residual energy** of the nodes.

$$ren1 \geq ren2 \geq \ldots.. \geq renk \tag{3}$$

This condition ensures the path nodes are ordered by decreasing residual energy.

$$min \; (rep1, rep2, rep3, \ldots, repn) \tag{4}$$

Where rep1, rep2,….,…, repn are the residual energies of different paths, and minimum selects the node with the lowest residual energy.

Finally, the destination node selects the maximum residual energy among all the paths by selecting the maximum value of the minimum energies:

$$max \, (min(rep1, rep2, rep3, \ldots, repn)) \tag{5}$$

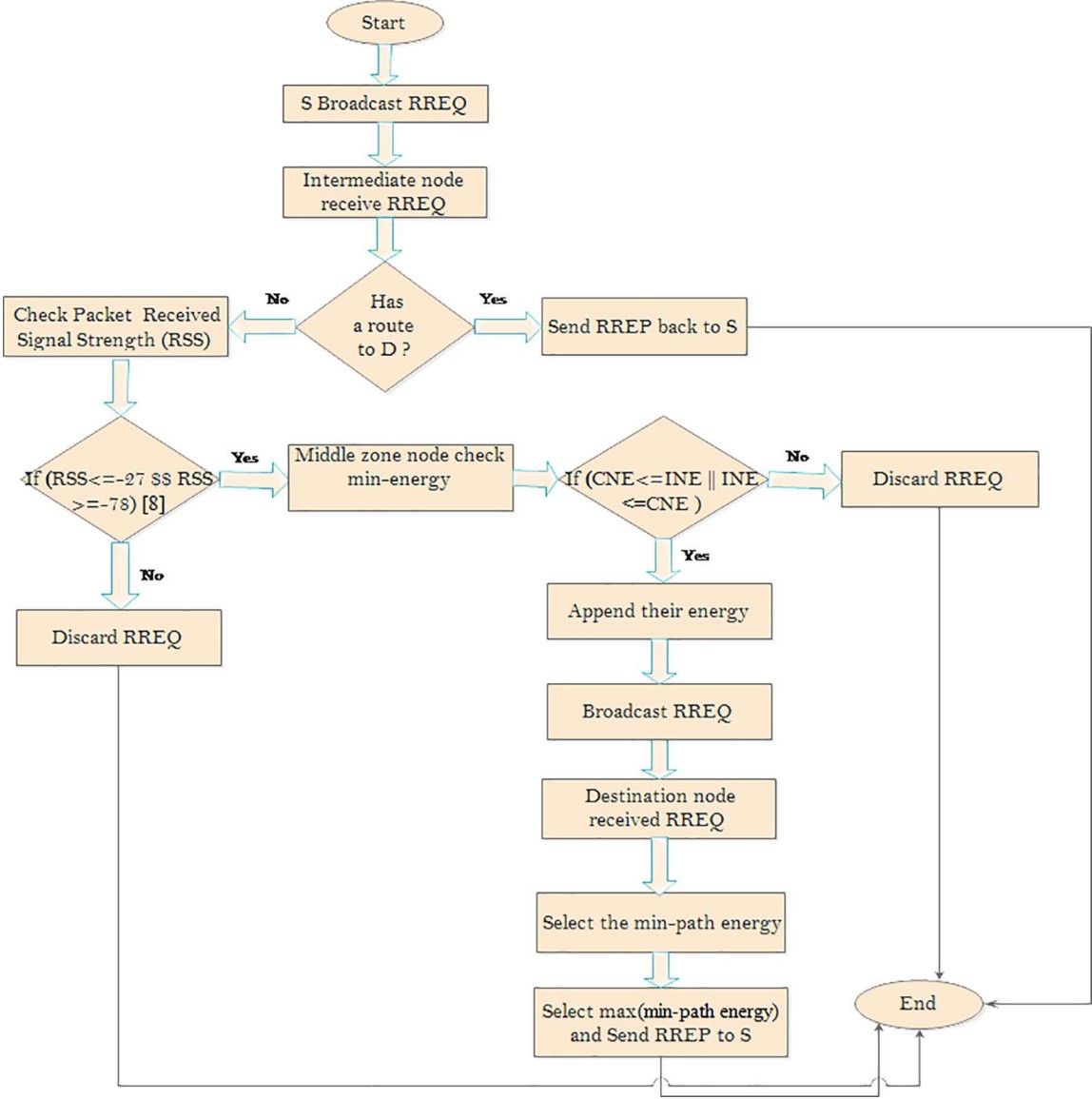

**Fig 3. Flowchart of proposed algorithm.**

Where Max represents the maximum energy value of the selected paths. Every time a new path is computed, the destination node sends a route reply with the maximum path energy.

The EASP-AODV algorithm enhances MANET longevity by integrating energy awareness into AODV, selecting energy-efficient routes, and dynamically adjusting paths based on real-time energy consumption. Simulated in NS-2, it demonstrated superior performance in packet delivery, energy efficiency, and network stability.

## 4. Results and discussion

This section compares the performance of the proposed Energy Aware Stable Path-AODV, Energy Aware Path Selection-AODV, and AODV. The performance is compared in terms of normalized routing overhead, packet delivery ratio, average end-to-end delay, normalized energy consumption, and network lifetime. NS2, which stands for Network Simulator version

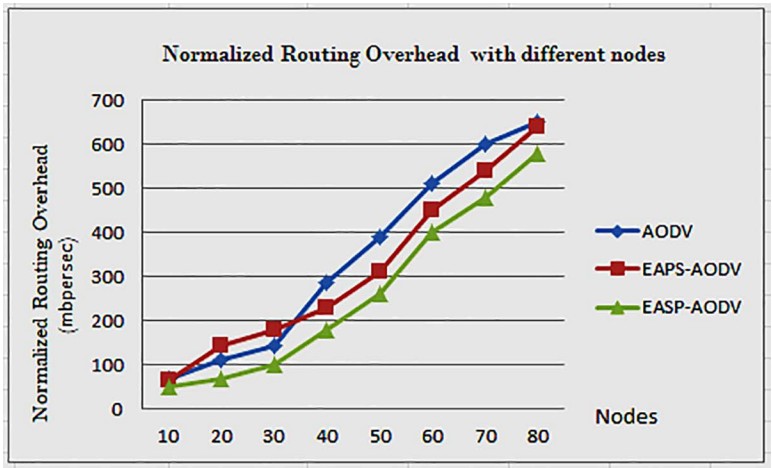

**Fig 4. Normalized routing overhead of AODV, EAPS-AODV, and EASP-AODV.**

**Table 2. AODV algorithm for route selection at node destination.**

| AODV algorithm for route selection at node destination |
| --- |
| DN = Destination Node |
| Min_eng = Minimum energy |
| Max (min-energy) = Maximum of minimum energy |
| RREP = Route Reply |
| If **DN** received the RREQ, then |
| Select min_energy ()// Destination node select the minimum energy among each path. |
| After selecting the min-eng of the path |
| Then, select Max (min-energy) |
| If the DN selects maximum path energy, then |
| Send RREP()// the DN selects the maximum path energy and sends RREP back to SN |
| Else |
| Discard RREQ()// Route Request is discarded |
| **End** |

2, was used to simulate this study. It is a simple programming tool known as an Event-driven simulation tool [28,29]. It is also effective and well-proven for dynamic network communication in nature. One of the criteria for selecting NS-2 is its flexibility in cooperating with wired and wireless protocols, such as routing algorithms, TCP, and UDP.

## 4.1. Results

### 4.1.1. Performance metrics.

A. End-to-End Delay

End-to-end delay (EED) is the time taken by a packet to go from source to destination divided by the number of data packets received. It is the amount of time it takes for a packet to travel from the source to the destination, averaged over the total number of data packets successfully received.

B. Packet Delivery Ratio

Packet delivery ratio (PDR) refers to the total number of packets which is received by the destination divided by the total number of packets which is sent by the source.

## C. Normalized Routing Overhead

Normalized routing overhead (NRL) refers to the ratio of the number of routing packets transmitted to the number of data packets successfully delivered to the destination. It provides a measure of the efficiency of the routing process relative to the actual data transmission. Control packet overhead is the amount of bandwidth and energy consumed by all routing protocol control packets. It helps in determining the scalability of a routing protocol. During route discovery, nodes generate more control packets, these packets consume more energy and create unstable MANET communication, to avoid this problem, the control packet is adjusted and reduced by limiting the number of nodes that are participating in packet flooding.

## D. Normalized Energy Consumption

Normalized Energy Consumption (NEC) measures network efficiency by calculating the total energy consumed by all nodes relative to the number of successfully received data packets. Since transmission and reception demand significant energy, NEC reflects the energy cost per delivered packet. Lower NEC indicates better energy efficiency and prolonged network lifespan. During packet transmission and reception, a large amount of energy is consumed, so normalized energy consumption means consumed energy per packet.

## E. Network Lifetime

In MANETs, network lifetime is constrained by the finite energy of nodes, making efficient energy management crucial. When a node with low energy receives a route request (RREQ), it may drop the packet, leading to higher delays and reduced packet delivery. Network lifetime is defined as the duration until nodes deplete their energy and can no longer function. Path life is no longer created in wireless networks, particularly MANETs, due to limited node energy. If one node broadcasts RREQ and the receiver node lacks efficient energy, the packet is dropped due to connection breakage, which increases end-to-end delay and decreases the packet delivery ratio. These types of issues reduce network lifetime, so considering single node energy has a high value in extending network lifetime. In short, network lifetime is the amount of time that a single node drains its energy and ceases to participate in network operation.

**4.1.2. Normalized routing overhead.** In Fig 4, before processing the RREQ, the intermediate node checks the incoming packet signal strength value. This limits the inner zone node and outer zone node, which helps to avoid the number of hops and reduce broken links. The routing overhead decreases as control packet flooding is limited.

When the number of node value is 10, the routing overhead is less, but when the number of nodes is 20, it adds some instant on creates overhead, as you see the number of nodes increased from 10 to 20, 30, 40, 50, 60, 70, 80, the control overhead also increases. Different factors cause the control overhead to increase. After the source node floods the control packet, the other neighboring nodes receive and rebroadcast that packet, without the use of routing overhead reduction techniques, whereas EAPS-AODV uses received signal strength for control packet reduction, but it limits the RREQ route participating nodes, but EAPS-AODV does not reduce destination overhead, which means the destination node selects the minimum residual energy from all appending nodes. The EASP-AODV algorithm we propose includes some functionality that helps to reduce destination node work overhead by comparing their energy before appending. As a result, each intermediate node compares its residual energy with the incoming node energy and then sends the minimum energy value rather than appending its residual energy without comparing, which results in destination overhead, because the destination node selects the minimum node energy after all nodes have reached it, so EASP-AODV compares before appending the energy, which reduces overhead. Based on this set of minimum and maximum techniques, the EASP-AODV algorithm selects the best

path with the highest path energy among the path replies and sends it. It improves network stability and increases network lifetime. In general, the EASP-AODV algorithm sends more data packets than the EAPS-AODV and AODV algorithms.

**4.1.3. Packet delivery ratio.** The packet delivery ratio is the ratio of the received packets to sent packets. According to the simulation result, Fig 5 shows the packet delivery ratio varies based on the number of nodes and the packet they transmit.

When the number of nodes increases, the packet delivery ratio also increases and sometimes decreases, due to the overhead. It suggested that the packet deliver ratio is increased in the proposed work because before the packet is forwarded, the incoming packet is checked with residual energy comparison on its routing table, so this minimum residual energy comparison limits the node. According to the number of nodes, for example at 30 nodes EASP-AODV performs 86.2%, and this shows the proposed algorithm limits the number of nodes that are participating in route discovery and route selection.

In general, the Packet Delivery Ratio confirms the ratio of packets successfully delivered to the destination versus packets sent to the destination by the source node (PDR). We can conclude from Fig 5 that the proposed algorithm has fewer packet drops than AODV and EAPS-AODV. As a result of the proposed work, an adequate amount of energy is saved, the network path remains stable and available for an extended period, and more packets are delivered to the destination node.

**4.1.4. Normalized energy consumption.** Normalized energy consumption (NEC) is expressed in energy consumption per packet. As shown in Fig 6, the energy consumption increases, are related to the number of nodes they are participating in the network operation, the number of control packets also increases because each node broadcasts the RREQ. This routing process consumes more energy.

It does not have some overhead reduction parameters all nodes are participating in the route finding or RREQ flooding, which leads to overhead. But in EAPS-AODV and EASP-AODV limited number of node participation, and however limited number of nodes are involved in packet flooding, EAPS-AODV does not limit the intermediate node and the destination node workload because that leads to the use of more energy, but EASP-AODV limits the participate nodes and reduce destination node energy consumption, how it could be done? Before appending their residual energy, first, the individual nodes compared their residual

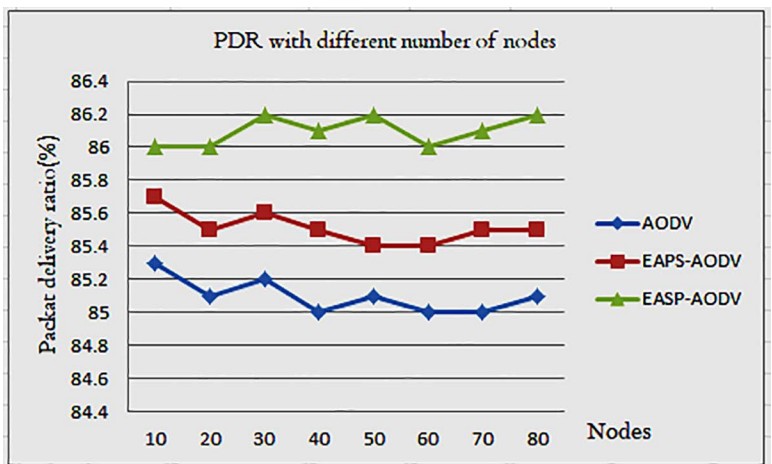

**Fig 5. Packet delivery ratio of AODV, EAPS-AODV, and EASP-AODV.**

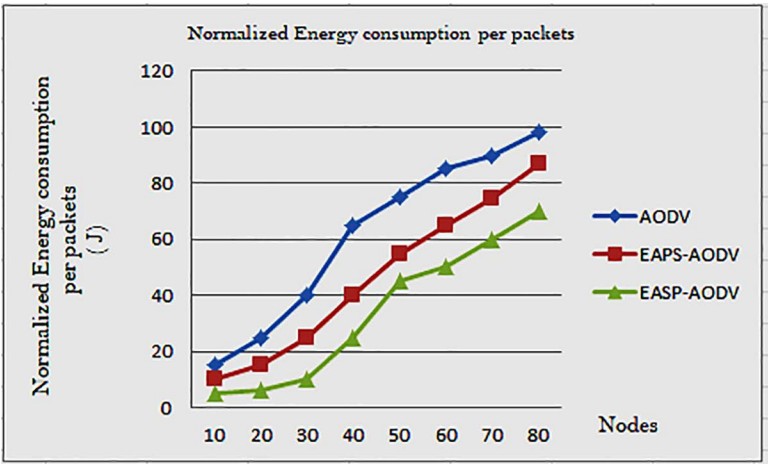

**Fig 6. Normalized energy consumption of AODV, EAPS-AODV, and EASP-AODV.**

energy with the incoming node residual energy, they compared the minimum residual energy with their neighbor node residual energy values, so after checking nodes that have minimum residual energy can flood the RREQ towards the destination node. So, the destination node task load and energy consumption are reduced.

When the nodes generate more RREQ, this leads to more consumption of energy. Based on the middle node route selection and the highest energy path selection, the number of nodes is limited. Even when the number of node energy consumption is increased, the EASP-AODV routing algorithm performs better for normalized energy consumption. Hence, the proposed routing algorithm reduces the energy consumption when EAPS-AODV and AODV are compared.

**4.1.5. Network lifetime.** According to Fig 7, the network lifetime decreases. This is because every node broadcasts packets. Even when they do not broadcast, the node consumes energy to check who is responsible for processing the control packet or RREQ.

For example, for the AODV, EAPS-AODV, and EASP-AODV routing algorithms, the network lifetime is reduced in nodes 10, 20, and 30. This is because more control packets are generated, which means that the amount of generated packets is less in node 10, consuming less energy, but high control packets are broadcasted in node 20. Each node in the transmission range consumes node energy by listening to the incoming packet. However, EASP-AODV limits the number of nodes that flood RREQ and selects the highest energy path by taking individual node residual energy into account. Therefore, the proposed method, EASP-AODV outperforms the EAPS-AODV and AODV routing algorithms concerning their network lifetime values. AE2E is the amount of time to transmit a data packet, it is defined as the packet received time (PRT) minus packet sending time (PST) over the received packet. In the network, delay occurs as a result of network size, network interference, congestion, and others.

Fig 8 shows that as the number of nodes increases, so does the average end-to-end delay. The simulation is performed with a variable number of nodes and at the same speed. When packet flooding is limited by reducing the number of nodes in the network, the amount of delay time is reduced. The EASP-AODV algorithm achieves a lower delay, as shown in Fig 8.

The network is not congested by the nodes, and packet flooding is sent by the chosen nodes, while the middle nodes only send the RREQ by choosing the highest energy path.

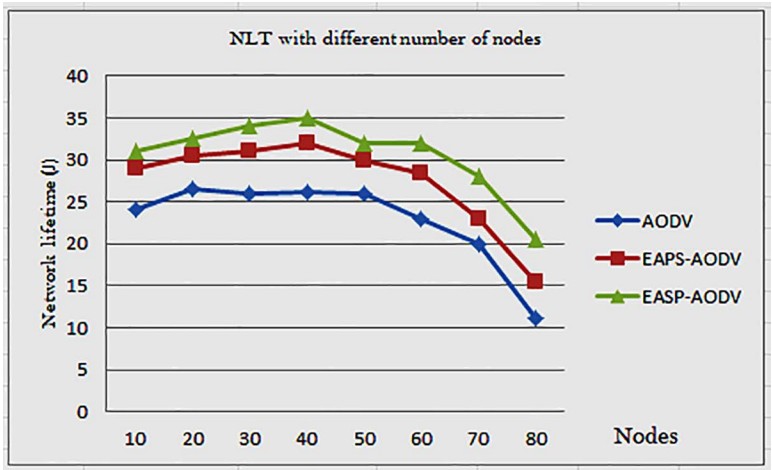

**Fig 7. Network lifetime of AODV, EAPS-AODV, and EASP-AODV.**

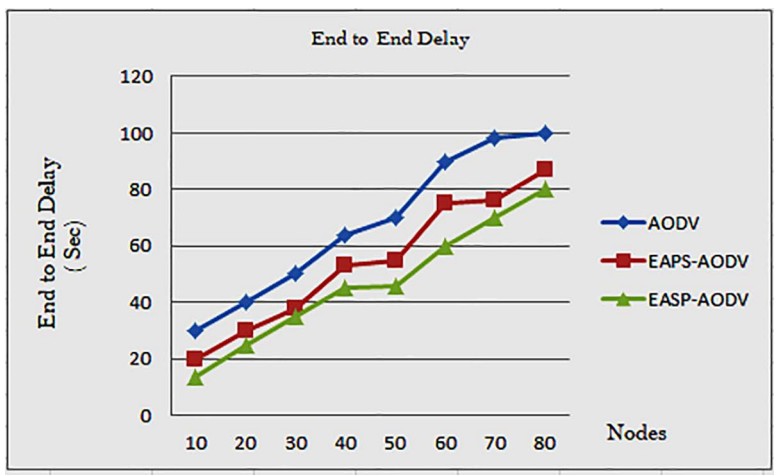

**Fig 8. Average end-to-end delay of AODV, EAPS-AODV, and EASP-AODV.**

When the number of nodes increases, so does the average end-to-end delay. This is due to node participation in the network, which is a fact we are investigating. But even at this, EASP-AODV outperforms the compared algorithms.

Therefore, decreasing average end-to-end delay means improving network performance and reducing link failure. As a result, the proposed EASP-AODV achieves some delay reduction. When compared to EAPS-AODV and AODV, EASP-AODV achieves less average end-to-end delay at node 10 and less delay at node 20. End-to-end delay is reduced because fewer nodes are congesting the network to send data packets from source to destination. End-to-end delay is reduced in general with limited control packets.

## 4.2. Discussion

In this study, we proposed the Energy-Aware Stable Path Ad hoc On-Demand Distance Vector (EASP-AODV) algorithm to extend the lifetime of Mobile Ad hoc Networks (MANETs) by integrating energy-awareness into the route discovery and maintenance process. By

selecting routes that prioritize nodes with sufficient energy, EASP-AODV minimizes the likelihood of route failures due to energy depletion, thus improving overall network stability and lifetime.

The method was implemented and evaluated using Network Simulator 2 (NS-2), which allowed us to simulate both energy consumption and mobility in a realistic MANET environment. **NS-2**'s flexibility in modeling routing protocols, coupled with the ability to customize energy models, made it an ideal choice for this research. We extended the standard AODV protocol within NS-2 to incorporate energy-aware features, such as energy tracking during transmission, reception, and idle states, as well as the dynamic selection of energy-efficient paths.

The results from our simulations directly reflect the effectiveness of the proposed method. **EASP-AODV** demonstrated significant improvements over standard AODV in terms of **packet delivery ratio (PDR), energy consumption,** and **network lifetime.** For instance, by avoiding paths through low-energy nodes, **EASP-AODV** was able to maintain a higher PDR, ensuring that data packets were successfully delivered without frequent route failures. This improvement in stability was directly linked to the energy-aware routing decisions implemented in the method. Additionally, EASP-AODV reduced overall energy consumption by balancing the load across nodes and preventing overuse of nodes with low energy, thereby prolonging the network's operational lifetime.

However, the simulation also highlighted some challenges, such as the scalability of the algorithm in large networks and the additional overhead from energy-related information exchange. These issues are a result of the dynamic energy monitoring and path recalculation required in EASP-AODV. While NS-2 provided a detailed platform for evaluation, the computational cost increases as the network size grows, this could affect the algorithm's performance in larger-scale real-world deployments.

In conclusion, the positive results obtained in terms of network stability and energy efficiency demonstrate the effectiveness of the **EASP-AODV** algorithm in addressing the energy limitations of traditional AODV. The use of NS-2 allowed us to closely examine the algorithm's impact on network performance and identify areas for improvement, such as optimizing scalability and refining energy models for more realistic simulations. Future work will focus on further enhancing the algorithm's efficiency in large networks and implementing more complex energy models to reflect real-world conditions more accurately. This study proposed the EASP-AODV algorithm to enhance MANET lifetime by integrating energy awareness into route discovery and maintenance. Simulated in NS-2, EASP-AODV outperformed AODV in packet delivery, energy efficiency, and network stability by prioritizing energy-rich nodes and balancing load distribution. Future work will focus on optimizing scalability and refining energy models for real-world applicability.

It became popular and well-known for research due to its flexible and modular nature [30–34].

**4.2.1. Simulation parameters.**  The EASP-AODV algorithm was evaluated using NS2, a widely used discrete-event simulator for modeling network protocols, including MANETs. NS2 enabled the simulation of network topologies and performance analysis based on metrics like packet delivery ratio, end-to-end delay, and normalized energy consumption. The results compared EASP-AODV with traditional AODV, highlighting its efficiency and stability. Table 3 lists various types of parameters used in this work. These parameters include a window XP with Cygwin for NS2 installation, a simulation area that covers 1000m x 1000m, a simulation time of 100 seconds, the number of nodes such as 10, 20, 30, 40, 50, 60, 70, and 80, initial energy of 100J. The type of channel is wireless because we are simulating a wireless mobile ad hoc node, the transmission range is 250 meters, the node mobility speed is 5–40ms, the packet

**Table 3. Simulation parameters.**

| Parameters | Values |
| --- | --- |
| Operating System | Window XP for Cygwin |
| Simulation Area | 1000m X 1000m |
| Simulation Time | 100s |
| Number of Nodes | 10, 20, 30, 40, 50, 60, 70 and 80 nodes |
| Energy Set up | Energy Model |
| Initial Node Energy | 100J |
| Channel Type | Wireless Channel |
| Transmission Range | 250 meters |
| Packet Size | 512 bytes |
| MAC Protocol | IEEE 802.11 |
| Antenna Model | Omni-directional |
| Mobility Model | Random Waypoint |
| Traffic Type | CBR |
| Routing protocol | AODV, EASP-AODV, EAPS-AODV |
| The upper threshold value for the inner zone | -27 dBm |
| The lower threshold value for the outer zone | -78 dBm |
| Traffic Connection | UDP |

size the node can forward is 512 bytes, the types of Media Access Control (MAC) protocol is IEEE 802.11n. 802.11 maintains node communication, so 802.11n is the new wireless standard component of Mac 802.11, ''n'' indicates the latest amendment of wireless local area network for high throughput of MANETs.

Because MANET nodes are dynamic, the antenna model used by the proposed algorithm in this work is omnidirectional. Because the direction of the mobile ad hoc network is undefined, broadcasting to all means that once the packet is broadcasted, it propagates in all directions around 360 degrees. For the signal propagation model, there are two-ray ground propagation models; this two-ray reflection model considers both the direct path and the ground reflection path. Omni (all-direction) antenna is used to include all participating nodes. Assume the source node wishes to send data to the destination node; at that time, the node initiates and broadcasts the route request to all neighbor nodes. This is because the source node does not know where the destination is, it could be on the right, left, in front, or the back, the RREQ is broadcasted to all directions, and the nearest node can receive it. Bidirectional antennas are better for efficient energy usage, but omnidirectional antennas are better for including all nodes by rotating 360 nodes. From these nodes, a high amount of node energy will be achieved which is not limited to forwarding RREQ and replying through RREP. The mobility model is of the random waypoint variety. Most researchers use Random waypoint (RWP), which is widely used for mobility modeling in MANET [35,36]. It has its parameters, which means that the node's starting time, destination (x and y values), and speed (how fast the node moves) are specified in the model. Constant bit rate (CBR) traffic means that the simulation bit rate is constant and supports audio and video communication.

The compared routing protocols are AODV, EASP-AODV, and EAPS-AODV. The ad hoc on-demand routing protocol has no energy-based parameters, but the existing energy-aware path selection algorithm considers the energy parameter, which is the average residual energy of the node, to increase the lifetime of the network of MANETs and discount link interruptions. The energy-aware AODV algorithm for stable paths does not consider the average residual energy of the nodes, but it considers the minimum residual energy of

the nodes and the number of hops for the selection of stable paths. These three algorithms are therefore simulated and their performance was compared as shown in Fig 7. During route discovery, the middle zone node is selected with two signal strength thresholds, the upper thresholds for the inner zone being $-27$ dBm and the lower thresholds for the outer zone being $-78$ dBm. But these thresholds are not fixed and can be changed to $-30$dBm and $-80$dBm. The results show that the signal intensity closest to the source node is high, and when the source node is far from the intermediate node, the signal intensity becomes low, so the threshold just describes the purpose of node classification of the inner zone, the middle zone and the outer node zone. After the signal strength is verified by the lower and upper threshold values, then the middle node broadcasts the packet, else the packet is discarded. The types of traffic connection used are TCP/UDP which is a transmission control protocol or user datagram protocol found in the transport layer. In this case, the main purpose of TCP/UDP is packet transmission. So, in our simulation we used UDP.

## 5. Conclusion and future work

### 5.1. Conclusion

Mobile ad hoc networks are decentralized networks and one type of infrastructure-less wireless network. By nature, the network has dynamic behavior, nodes are self-configured and they are self-controlled. Like wired networks, MANETS does not have a centralized administration mechanism, this creates link breakage caused by node mobility and limited node energy. It affects the network lifetime and the overall network performance. In this paper, an EASP-AODV algorithm was developed by using individual nodes with minimal residual energy and received signal strength. To demonstrate the performance of the proposed algorithm, we performed a performance analysis by using the NS2 simulation tool. The **EASP-AODV algorithm** reduces link breakage in **MANETs** by prioritizing nodes with higher residual energy, ensuring stable routes. It incorporates energy-aware path selection and sets energy thresholds to avoid nodes with low energy, which reduces the likelihood of link failures. The algorithm dynamically adjusts routes based on energy levels and node stability, preventing sudden link breakages. In case of failure, it employs localized route recovery to quickly find alternative paths. These measures help maintain stable communication and minimize link disruptions. So, the main target of this research is extending the network lifetime of MANETs. According to the simulation results, the proposed algorithm outperforms the AODV and EAPS-AODV routing algorithms in terms of average end-to-end delay, packet delivery ratio, normalized routing overhead, network lifetime, and normalized energy consumption. The simulation results show that the normalized energy consumption of EASP-AODV is 4.5% and 9.5% less than Energy Aware Path Selection (EAPS-AODV) and AODV routing algorithms, and it prolongs the network lifetime by 3.5% and 7.5% than EAPS-AODV and AODV routing algorithms, respectively.

### 5.2. Future work

EASP-AODV extends MANET lifetime by prioritizing energy-efficient routes, avoiding low-energy nodes, and proactively maintaining stability. It reduces link failures, minimizes control overhead, and balances energy consumption across the network. These strategies enhance reliability and ensure prolonged network performance In the future, authors hope to add node location consideration with the received signal strength which would further improve the network lifetime.

## Author contributions

**Conceptualization:** Tibebu Legesse, Dagne Walle Girmaw.

**Data curation:** Tibebu Legesse, Dagne Walle Girmaw.

**Formal analysis:** Esubalew Yitayal.

**Investigation:** Tibebu Legesse, Dagne Walle Girmaw, Esubalew Yitayal.

**Methodology:** Tibebu Legesse, Dagne Walle Girmaw.

**Resources:** Tibebu Legesse, Engida Admassu.

**Software:** Engida Admassu.

**Supervision:** Esubalew Yitayal, Engida Admassu.

**Validation:** Tibebu Legesse, Dagne Walle Girmaw, Engida Admassu.

**Visualization:** Tibebu Legesse.

**Writing – original draft:** Tibebu Legesse.

**Writing – review & editing:** Esubalew Yitayal.

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
