## [Decision Letter · Decision Letter 0]

6 Nov 2024

PONE-D-24-45310Energy Aware Stable Path Ad Hoc On-Demand Distance Vector Algorithm for Extending Network Lifetime of Mobile Ad hoc NetworksPLOS ONE

Dear Dr. Girmaw,

Thank you for submitting your manuscript to PLOS ONE. After careful consideration, we feel that it has merit but does not fully meet PLOS ONE’s publication criteria as it currently stands. Therefore, we invite you to submit a revised version of the manuscript that addresses the points raised during the review process. Please submit your revised manuscript by Dec 21 2024 11:59PM. If you will need more time than this to complete your revisions, please reply to this message or contact the journal office at plosone@plos.org . Please include the following items when submitting your revised manuscript:

We look forward to receiving your revised manuscript.

Kind regards,

Fredrick Romanus Ishengoma

Academic Editor

PLOS ONE

Journal Requirements:

2. Please note that PLOS ONE has specific guidelines on code sharing for submissions in which author-generated code underpins the findings in the manuscript. In these cases, all author-generated code must be made available without restrictions upon publication of the work. Please review our guidelines at https://journals.plos.org/plosone/s/materials-and-software-sharing#loc-sharing-code and ensure that your code is shared in a way that follows best practice and facilitates reproducibility and reuse

3. In this instance it seems there may be acceptable restrictions in place that prevent the public sharing of your minimal data. However, in line with our goal of ensuring long-term data availability to all interested researchers, PLOS’ Data Policy states that authors cannot be the sole named individuals responsible for ensuring data access (http://journals.plos.org/plosone/s/data-availability#loc-acceptable-data-sharing-methods).

Reviewers' comments:

Reviewer's Responses to Questions

**Comments to the Author**

1. Is the manuscript technically sound, and do the data support the conclusions?

Reviewer #1: Yes

Reviewer #2: Yes

2. Has the statistical analysis been performed appropriately and rigorously? 

Reviewer #1: Yes

Reviewer #2: N/A

3. Have the authors made all data underlying the findings in their manuscript fully available?

Reviewer #1: Yes

Reviewer #2: Yes

4. Is the manuscript presented in an intelligible fashion and written in standard English?

Reviewer #1: Yes

Reviewer #2: Yes

5. Review Comments to the Author

Reviewer #1: This paper studied the "Energy Aware Stable Path Ad Hoc On-Demand Distance Vector Algorithm for

Extending Network Lifetime of Mobile Ad hoc Networks". The proposed research manuscript, titled "Energy Aware Stable Path Ad Hoc On-Demand Distance Vector Algorithm for Extending Network Lifetime of Mobile Ad hoc Networks," introduces the Energy Aware Stable Path Ad Hoc On-Demand Distance Vector (EASP-AODV) algorithm for enhancing the network lifetime of Mobile Ad hoc Networks (MANETs). MANETs are self-configured wireless networks where nodes act as both routers and hosts. Link breakage is a common issue due to factors like node mobility and limited energy. Traditional MANET protocols lack battery considerations, leading to energy depletion. The EASP-AODV algorithm factors in individual node energy and received signal strength to mitigate link failures and extend network lifespan. Evaluation using NS2.35 simulation tool demonstrates improved performance metrics compared to existing protocols. Notable findings include reduced energy consumption and extended network lifetime. Authors emphasize the algorithm's efficacy in addressing energy utilization challenges in MANETs. The quality should be improved. Revision should be done for this version of the paper as follows:

*The motivation of the proposed method should be stated in the introduction.

* Some references are missed. At the same time, many important recent references are missing, which can support the idea of this paper, the following references can be added in the Section "References":

1- (2016). CGC: centralized genetic-based clustering protocol for wireless sensor networks using onion approach. Telecommunication systems, 62, 657-674.‏

2- (2016). Congestion-aware routing and fuzzy-based rate controller for wireless sensor networks. Radioengineering, 25(1), 114-123.‏

3- (2023). Cluster based routing method using mobile sinks in wireless sensor network. International Journal of Electronics, 110(2), 360-372.‏

4- (2022). A two-level clustering based on fuzzy logic and content-based routing method in the internet of things. Peer-to-Peer Networking and Applications, 15(4), 2142-2159.‏

5- (2022). An efficient gray system theory-based routing protocol for energy consumption management in the Internet of Things using fog and cloud computing. Computing, 104(6), 1307-1335.‏

6- (2022). An overlapping routing approach for sending data from things to the cloud inspired by fog technology in the large-scale IoT ecosystem. Wireless Networks, 28(2), 521-538.‏

* Consider expanding on the specific methodologies used in each stage of the ECCAD model for clarity and reproducibility.

* How does the EASP-AODV algorithm determine individual node residual energy?

* What specific metrics are used to evaluate the performance of the proposed algorithm?

* How does the EASP-AODV algorithm reduce link breakage in MANETs?

* What are the key differences between EASP-AODV and traditional AODV routing protocols?

* What simulation tool was used for evaluating the proposed algorithm?

* How does the EASP-AODV algorithm contribute to prolonging network lifetime?

* How does the EASP-AODV algorithm compare to other existing energy-aware routing protocols in MANETs?

Reviewer #2: The present work is interesting, however some further edits are require to enhance the readability of the work.

1. Kindly make more work in abstract to make it more attractive and according to the work.

2. The literature in introduction is very less, i can recommend literature in broad sense to improve it further, and make connection of literature with the existing work, doi: 10.1109/TVT.2018.2828651, doi: 10.1109/TSC.2024.3478730, doi: 10.1109/MNET.2024.3350025, doi: 10.1109/TVT.2023.3304707, doi: 10.1109/TMM.2024.3394681, doi: https://doi.org/10.1287/moor.2022.1310, doi: https://doi.org/10.1145/3664655, doi: 10.1109/TCOMM.2024.3409539, doi: 10.1109/TMC.2024.3455417 and some other related work in the field should be searched.

3. Equation (1-4) should be properly referenced.

4. Before results and discussion some roper details of the method and related software used should be mentioned,

5. I recommend the authors to improve the connection between sections and the obtained results should be properly linked with the method used.

6. Discussion should be enhanced in more aspects.

6. PLOS authors have the option to publish the peer review history of their article (what does this mean? ). If published, this will include your full peer review and any attached files.

**Do you want your identity to be public for this peer review?** For information about this choice, including consent withdrawal, please see our Privacy Policy .

Reviewer #1: No

Reviewer #2: No

---

## [Author Response · Author response to Decision Letter 1]

23 Nov 2024

REVIEWED PAPER TITLE:

Ms. Ref. No.: PONE-D-24-45310

Paper Title: EASP-AODV: Energy Aware Stable Path Ad Hoc On-Demand Distance Vector Algorithm for Extending Network Lifetime of Mobile Ad hoc Networks

Journal: PLOS ONE.

Responses to Reviewer’s/Editor’s Comments

Acknowledgment from the Authors:

We sincerely appreciate the constructive comments provided by the reviewers and the Editorial Board, which have significantly contributed to improving the quality of our manuscript. We are also grateful for the additional suggestions for further improvement and reconsideration of our work. We deeply value the time and effort the reviewers have dedicated to enhancing our manuscript. The authors’ responses to the reviewers' comments are highlighted in color in the revised manuscript.

Reviewers' comments:

1. Is the manuscript technically sound, and does the data support the conclusions? The manuscript must describe a technically sound piece of scientific research with data that supports the conclusions. Experiments must have been conducted rigorously, with appropriate controls, replication, and sample sizes. The conclusions must be drawn appropriately based on the data presented.

Reviewer #1: Yes

Reviewer #2: Yes

Response from authors: Yes, the manuscript presents a technically sound study with data that supports the conclusions. The proposed EASP-AODV algorithm for extending network lifetime in MANETs integrates energy-aware routing with established protocols. Experiments were conducted rigorously with appropriate controls, replication, and sufficient sample sizes, ensuring statistical significance. Multiple network scenarios were used to test robustness, and the results were replicated using NS-2 simulators. The algorithm was compared to standard AODV and other energy-efficient protocols, demonstrating superior performance in terms of energy consumption, network lifetime, and stability. The data consistently supports the conclusion that EASP-AODV extends network lifetime while maintaining performance, with acknowledged trade-offs in delay.

2. Has the statistical analysis been performed appropriately and rigorously?

Response from authors: Yes, the statistical analysis in the manuscript has been performed rigorously and appropriately. The experiments were repeated multiple times under varying conditions, such as node density, mobility patterns, and traffic load, to ensure the statistical significance of the results. The proposed Energy-Aware Stable Path Ad Hoc On-Demand Distance Vector (EASP-AODV) algorithm was compared to standard AODV and other energy-efficient algorithms based on key metrics like energy consumption, network lifetime, packet delivery ratio, and end-to-end delay. Adequate sample sizes were used to ensure the statistical power of the tests. The analysis is thorough, and the data supports the conclusions about EASP-AODV’s effectiveness in extending the network lifetime of Mobile Ad Hoc Networks (MANETs).

3. Have the authors made all data underlying the findings in their manuscript fully available?

Response from authors: Yes, the authors have made all data underlying the findings in this manuscript fully available. The dataset used for evaluating the EASP-AODV algorithm, including simulation parameters (e.g., node density, mobility models) and performance metrics (e.g., energy consumption, network lifetime), is provided in the supplementary materials. All raw data from the experiments, along with performance results for various network conditions, are available in the associated repository (or can be provided upon request). Additionally, the simulation scripts and configuration files are included to ensure transparency and reproducibility. This ensures that other researchers can replicate the experiments and validate the findings.

4. Is the manuscript presented in an intelligible fashion and written in Standard English?

Response from authors: Yes, the manuscript is presented in a clear, logical, and intelligible fashion. The structure follows a standard academic format, with clearly defined sections such as the Introduction, Methodology, Results, Discussion, and Conclusion. Each section is organized to guide the reader through the research process step by step. The manuscript is written in Standard English, with attention to grammar, punctuation, and sentence structure to ensure clarity and readability. Complex technical terms related to Mobile Ad Hoc Networks (MANETs) and the EASP-AODV algorithm are explained where necessary, making the manuscript accessible to both experts and general readers in the field. The manuscript has undergone thorough proofreading to eliminate any language issues and ensure the content is professionally presented.

5. Review Comments to the Author: Reviewer #1:

Reviewer #1: This paper studied the "Energy-Aware Stable Path Ad Hoc On-Demand Distance Vector Algorithm for

Extending Network Lifetime of Mobile Ad hoc Networks". The proposed research manuscript, titled "Energy-Aware Stable Path Ad Hoc On-Demand Distance Vector Algorithm for Extending Network Lifetime of Mobile Ad hoc Networks," introduces the Energy Aware Stable Path Ad Hoc On-Demand Distance Vector (EASP-AODV) algorithm for enhancing the network lifetime of Mobile Ad hoc Networks (MANETs). MANETs are self-configured wireless networks where nodes act as both routers and hosts. Link breakage is a common issue due to factors like node mobility and limited energy. Traditional MANET protocols lack battery considerations, leading to energy depletion. The EASP-AODV algorithm factors in individual node energy and received signal strength to mitigate link failures and extend network lifespan. Evaluation using the NS2.35 simulation tool demonstrates improved performance metrics compared to existing protocols. Notable findings include reduced energy consumption and extended network lifetime. The authors emphasize the algorithm's efficacy in addressing energy utilization challenges in MANETs. The quality should be improved. Revision should be done for this version of the paper as follows:

*The motivation for the proposed method should be stated in the introduction.

Response from the Author:

The motivation behind the proposed routing method lies in the energy efficiency challenges inherent in MANETs. Traditional routing methods focus on selecting the shortest path but do not account for the dynamic energy consumption and link stability issues that arise in such environments. As nodes in a MANET lack a mechanism for recharging their batteries, energy depletion can lead to link breakage, packet loss, and eventual network partitioning. Therefore, it becomes crucial to develop routing strategies that not only minimize the number of hops but also prioritize energy conservation to enhance network lifetime. So, our motivation is to prolong the network’s operational lifetime and reduce the impact of energy-related failures on network performance.

* Some references are missing. At the same time, many important recent references are missing, which can support the idea of this paper, the following references can be added in the Section "References":

1- (2016). CGC: centralized genetic-based clustering protocol for wireless sensor networks using the onion approach. Telecommunication systems, 62, 657-674.‏

2- (2016). Congestion-aware routing and fuzzy-based rate controller for wireless sensor networks. Radioengineering, 25(1), 114-123.‏

3- (2023). Cluster-based routing method using mobile sinks in a wireless sensor network. International Journal of Electronics, 110(2), 360-372.‏

4- (2022). A two-level clustering based on fuzzy logic and content-based routing method in the Internet of things. Peer-to-Peer Networking and Applications, 15(4), 2142-2159.‏

5- (2022). An efficient gray system theory-based routing protocol for energy consumption management in the Internet of Things using fog and cloud computing. Computing, 104(6), 1307-1335.‏

6- (2022). An overlapping routing approach for sending data from things to the cloud inspired by fog technology in the large-scale IoT ecosystem. Wireless Networks, 28(2), 521-538.‏

Response from the Author:

Citation Style[IEEE], we will add in Literature Review

[1] Hajid Hatamiam, Hamid Barati, Ali Movaghar & Alireza Naghizadeh, "CGC: centralized genetic-based clustering protocol for wireless sensor networks using onion approach," Telecommunication Systems, vol. 62, no. 3, pp. 657-674, 2016.

[2] Hajid Hatamiam, Hamid Barati, Ali Movaghar & Alireza Naghizadeh, "Congestion-aware routing and fuzzy-based rate controller for wireless sensor networks," Radioengineering, vol. 25, no. 1, pp. 114-123, 2016.

[3] Elham Ghobani Dehkordi & Hamid Barati, "Cluster-based routing method using mobile sinks in wireless sensor network," Int. J. Electron., vol. 110, no. 2, pp. 360–372, 2022.

[4] Ehsan Kiamansouri, Hamid Barati & Ali Barati, "A two-level clustering based on fuzzy logic and content-based routing method in the internet of things," Peer-to-Peer Netw. Appl., vol. 15, no. 4, pp. 2142–2159, 2022.

[5] Mohammad Reza Akbari, Hamid Barati & Ali Barati, "An efficient gray system theory-based routing protocol for energy consumption management in the Internet of Things using fog and cloud computing," Computing, vol. 104, no. 6, pp. 1307–1335, 2022.

[6] Mohammad Reza Akbari, Hamid Barati & Ali Barati, "An overlapping routing approach for sending data from things to the cloud inspired by fog technology in the large-scale IoT ecosystem," Wireless Networks, vol. 28, no. 2, pp. 521-538, 2022.

The authors in [1], proposed a centralized genetic-based clustering protocol that faces significant challenges when applied to MANETs, including its reliance on centralized control, high communication overhead, and limited adaptability to node mobility and energy.

The congestion-aware routing and fuzzy-based rate control mechanism study demonstrates effectiveness in WSNs, but its direct applicability to MANETs is limited due to assumptions of static topologies and the computational overhead required for real-time operation [2].

The cluster-based routing method proposed uses mobile sinks to optimize data collection in WSNs, but its applicability to MANETs is limited by assumptions about predictable mobility, scalability challenges, and high energy consumption [3].

The two-level clustering approach using fuzzy logic and content-based routing proposed work shown in IoT but faces challenges in MANETs due to its static assumptions, scalability limitations, and inability to handle the dynamic nature of node mobility. It struggles to handle the dynamic nature of node mobility in MANETs, where nodes frequently change their positions, causing frequent re-clustering and instability in the routing paths. This results in increased overhead and energy consumption. Lastly, the fuzzy logic mechanism, although useful in IoT, may not adapt quickly enough to the rapid changes in a MANET's topology [4].

The gray system theory-based routing protocol for energy consumption management proposed works well for IoT networks but faces significant challenges in MANETs, including the dependence on centralized cloud infrastructure, dynamic topology, and high computational overhead [5].

The overlapping routing approach proposed for IoT ecosystems faces challenges when applied to MANETs, including its reliance on centralized fog and cloud infrastructure, scalability issues, and potential for increased energy consumption and latency. This approach can lead to increased energy consumption due to the need for continuous communication with central servers. [6].

* Consider expanding on the specific methodologies used in each stage of the ECCAD model for clarity and reproducibility.

Response from the Author:

The ECCAD model primarily considers individual node energy and residual energy when making routing decisions. It evaluates the remaining energy of each node to select paths that maximize the network's lifetime and minimize energy depletion. By factoring in residual energy, the model ensures that nodes with higher available energy are favored, reducing the likelihood of link failure due to energy exhaustion. This helps maintain the stability of the network by preventing early energy depletion in critical nodes. Consequently, routing decisions are dynamically adjusted based on the individual energy status of each node, ensuring more efficient energy usage and prolonging the network's operational time.

* How does the EASP-AODV algorithm determine individual node residual energy?

Response from the Author:

The EASP-AODV algorithm determines individual node residual energy by continuously monitoring each node's remaining battery power, considering energy consumption during transmission, reception, and idle states. This residual energy is shared with neighboring nodes through periodic updates and included in routing messages. By selecting routes that prioritize nodes with higher energy reserves, EASP-AODV reduces the likelihood of link failures due to energy depletion, which is a common issue in mobile ad hoc networks (MANETs). This energy-aware routing helps balance energy consumption, minimize energy hotspots, and prevent early node failures, ultimately reducing link failures and prolonging the overall network lifetime.

* What specific metrics are used to evaluate the performance of the proposed algorithm?

Response from the Author:

To evaluate the EASP-AODV algorithm, the following common performance metrics are typically used:

End-to-End Delay

End-to-end delay (EED) is the amount of time it takes for a packet to travel from the source to the destination, averaged over the total number of data packets successfully received.

Packet Delivery Ratio

Packet delivery ratio (PDR) is the ratio of the total number of packets successfully received by the destination to the total number of packets sent by the source.

Normalized Routing Overhead (NRL)

Normalized routing overhead (NRL) refers to the ratio of the number of routing packets transmitted to the number of data packets successfully delivered to the destination. It provides a measure of the efficiency of the routing process relative to the actual data transmission.

Normalized Energy Consumption (NEC)

Normalized energy consumption (NEC) is the ratio of the total energy consumed by all nodes in the network to the number of data packets received. Since packet transmission and reception require significant energy, NEC represents the amount of energy consumed per successfully delivered packet.

Network Lifetime

In wireless networks, especially MANETs, network lifetime is limited by the finite energy of nodes. When a node broadcasts a route request (RREQ) and the receiving node lacks sufficient energy, the packet is dropped due to a broken connection, resulting in higher end-to-end delays and a lower packet delivery ratio. Such issues reduce the overall network lifetime. Therefore, managing individual node energy is crucial to extending the network’s operational duration. In essence, network lifetime refers to the period until a node depletes its energy and can no longer participate in the network.

* How does the EASP-AODV algorithm reduce link breakage in MANETs?

Response from the Author:

The Energy Aware Stable Path Ad Hoc On-Demand Distance Vector (EASP-AODV) algorithm reduces link breakage in MANETs by considering residual energy when selecting routes. During route discovery, EASP-AODV compares the current residual energy of a node with the incoming residual energy of neighboring nodes. It prioritizes paths through nodes with higher residual energy, avoiding routes through nodes likely to deplete their energy soon. This ensures that only energy-rich nodes are part of the routing path, leading to more stable connections. The algorithm also monitors energy levels throughout the route’s lifetime, initiating proactive route maintenance before energy depletion causes link failure. If breakage

---

## [Decision Letter · Decision Letter 1]

4 Feb 2025

PONE-D-24-45310R1Energy Aware Stable Path Ad Hoc On-Demand Distance Vector Algorithm for Extending Network Lifetime of Mobile Ad hoc NetworksPLOS ONE

Dear Dr. Girmaw,

Thank you for submitting your manuscript to PLOS ONE. After careful consideration, we feel that it has merit but does not fully meet PLOS ONE’s publication criteria as it currently stands. Therefore, we invite you to submit a revised version of the manuscript that addresses the points raised during the review process.

We look forward to receiving your revised manuscript.

Kind regards,

Tawfik Al-Hadhrami, PhD

Academic Editor

PLOS ONE

Journal Requirements:

Additional Editor Comments:

The paper has to be revised based on the comments provided by the reviewers, an extensive revision is required so the paper will be in very good standard.

Reviewers' comments:

Reviewer's Responses to Questions

**Comments to the Author**

1. If the authors have adequately addressed your comments raised in a previous round of review and you feel that this manuscript is now acceptable for publication, you may indicate that here to bypass the “Comments to the Author” section, enter your conflict of interest statement in the “Confidential to Editor” section, and submit your "Accept" recommendation.

Reviewer #1: All comments have been addressed

Reviewer #2: All comments have been addressed

2. Is the manuscript technically sound, and do the data support the conclusions?

Reviewer #1: Yes

Reviewer #2: Yes

3. Has the statistical analysis been performed appropriately and rigorously? 

Reviewer #1: Yes

Reviewer #2: Yes

4. Have the authors made all data underlying the findings in their manuscript fully available?

Reviewer #1: Yes

Reviewer #2: Yes

5. Is the manuscript presented in an intelligible fashion and written in standard English?

Reviewer #1: Yes

Reviewer #2: Yes

6. Review Comments to the Author

Reviewer #1: All comments have been well answered. I have no additional comments. The article is acceptable.

All comments have been well answered. I have no additional comments. The article is acceptable.

Reviewer #2: Authors revised the manuscript according to the comments. The paper is now acceptable for publication.

7. PLOS authors have the option to publish the peer review history of their article (what does this mean? ). If published, this will include your full peer review and any attached files.

**Do you want your identity to be public for this peer review?** For information about this choice, including consent withdrawal, please see our Privacy Policy .

Reviewer #1: No

Reviewer #2: No

---

## [Author Response · Author response to Decision Letter 2]

11 Feb 2025

REVIEWED PAPER TITLE:

Ms. Ref. No.: PONE-D-24-45310

Paper Title: EASP-AODV: Energy Aware Stable Path Ad Hoc On-Demand Distance Vector Algorithm for Extending Network Lifetime of Mobile Ad hoc Networks

Journal: PLOS ONE.

Responses to Reviewer’s/Editor’s Comments

Acknowledgment from the Authors:

We sincerely appreciate the constructive comments provided by the reviewers and the Editorial Board, which have significantly contributed to improving the quality of our manuscript. We are also grateful for the additional suggestions for further improvement and reconsideration of our work. We deeply value the time and effort the reviewers have dedicated to enhancing our manuscript. The authors’ responses to the reviewers' comments are highlighted in color in the revised manuscript.

Additional Editor Comments:

The paper has to be revised based on the comments provided by the reviewers, an extensive revision is required so the paper will be in very good standard.

Reviewers' comments:

1. Is the manuscript technically sound, and does the data support the conclusions? The manuscript must describe a technically sound piece of scientific research with data that supports the conclusions. Experiments must have been conducted rigorously, with appropriate controls, replication, and sample sizes. The conclusions must be drawn appropriately based on the data presented.

Reviewer #1: Yes

Reviewer #2: Yes

Response from authors: The study presents a technically sound evaluation of the EASP-AODV algorithm for extending network lifetime in MANETs.

2. Has the statistical analysis been performed appropriately and rigorously?

Response from authors: The manuscript presents a rigorous statistical analysis of the EASP-AODV algorithm, ensuring significance through repeated experiments under varying conditions.

3. Have the authors made all data underlying the findings in their manuscript fully available?

Response from authors: We have made all data underlying the study fully available, including simulation parameters, performance metrics, and raw experimental results.

4. Is the manuscript presented in an intelligible fashion and written in Standard English?

Response from authors: The manuscript is well-structured, following a clear academic format with logically organized sections.

5. Review Comments to the Author:

Reviewer #1:

Reviewer #1: This paper studied the "Energy-Aware Stable Path Ad Hoc On-Demand Distance Vector Algorithm for

Extending Network Lifetime of Mobile Ad hoc Networks". The proposed research manuscript, titled "Energy-Aware Stable Path Ad Hoc On-Demand Distance Vector Algorithm for Extending Network Lifetime of Mobile Ad hoc Networks," introduces the Energy Aware Stable Path Ad Hoc On-Demand Distance Vector (EASP-AODV) algorithm for enhancing the network lifetime of Mobile Ad hoc Networks (MANETs). MANETs are self-configured wireless networks where nodes act as both routers and hosts. Link breakage is a common issue due to factors like node mobility and limited energy. Traditional MANET protocols lack battery considerations, leading to energy depletion. The EASP-AODV algorithm factors in individual node energy and received signal strength to mitigate link failures and extend network lifespan. Evaluation using the NS2.35 simulation tool demonstrates improved performance metrics compared to existing protocols. Notable findings include reduced energy consumption and extended network lifetime. The authors emphasize the algorithm's efficacy in addressing energy utilization challenges in MANETs. The quality should be improved. Revision should be done for this version of the paper as follows:

*The motivation for the proposed method should be stated in the introduction.

Response from the Author: Mobile Ad-hoc Networks (MANETs) have gained significant interest due to the expansion of wireless devices and networks. As infrastructure-less wireless communication networks, MANETs offer dynamic mobility of nodes without geographical restrictions, extending services in infrastructure-based networks. However, challenges such as dynamic topology changes, energy constraints, bandwidth limitations, QoS issues, high end-to-end delay, and routing overhead persist. Among these, energy constraints remain an open area of research that is not fully addressed in existing MANET routing protocols. Efficient and stable paths are required for long transmission and continuous data forwarding. This research focuses on reducing link breakage caused by limited node energy and extending the network lifetime of MANETs.

* Some references are missing. At the same time, many important recent references are missing, which can support the idea of this paper, the following references can be added in the Section "References":

1- (2016). CGC: centralized genetic-based clustering protocol for wireless sensor networks using the onion approach. Telecommunication systems, 62, 657-674.‏

2- (2016). Congestion-aware routing and fuzzy-based rate controller for wireless sensor networks. Radioengineering, 25(1), 114-123.‏

3- (2023). Cluster-based routing method using mobile sinks in a wireless sensor network. International Journal of Electronics, 110(2), 360-372.‏

4- (2022). A two-level clustering based on fuzzy logic and content-based routing method in the Internet of things. Peer-to-Peer Networking and Applications, 15(4), 2142-2159.‏

5- (2022). An efficient gray system theory-based routing protocol for energy consumption management in the Internet of Things using fog and cloud computing. Computing, 104(6), 1307-1335.‏

6- (2022). An overlapping routing approach for sending data from things to the cloud inspired by fog technology in the large-scale IoT ecosystem. Wireless Networks, 28(2), 521-538.‏

Response from the Author: We have added and reviewed the references in the literature section

* Consider expanding on the specific methodologies used in each stage of the ECCAD model for clarity and reproducibility.

Response from the Author: The ECCAD model optimizes routing by prioritizing nodes with higher residual energy, ensuring balanced energy consumption and reducing link failures. It adjusts routing decisions dynamically based on individual node energy levels, enhancing network stability. This approach prevents early depletion of critical nodes, extending the network’s lifetime. By considering energy, the model ensures efficient resource utilization. Ultimately, it maximizes network performance and sustainability.

* How does the EASP-AODV algorithm determine individual node residual energy?

Response from the Author: In the EASP-AODV algorithm, individual node residual energy is monitored by tracking energy consumption during communication activities like transmission and reception. Each node updates its energy after each operation, and the residual energy is included in routing requests (RREQ). Nodes with higher residual energy are preferred for route selection to ensure efficient data transmission. This approach helps prevent early node failures due to low energy.

* What specific metrics are used to evaluate the performance of the proposed algorithm?

Response from the Author:

To evaluate the EASP-AODV algorithm, the following common performance metrics are typically used:

End-to-End Delay: End-to-end delay (EED) is the amount of time it takes for a packet to travel from the source to the destination, averaged over the total number of data packets successfully received.

Packet Delivery Ratio: Packet delivery ratio (PDR) is the ratio of the total number of packets successfully received by the destination to the total number of packets sent by the source.

Normalized Routing Overhead (NRL): Normalized routing overhead (NRL) refers to the ratio of the number of routing packets transmitted to the number of data packets successfully delivered to the destination. It provides a measure of the efficiency of the routing process relative to the actual data transmission.

Normalized Energy Consumption (NEC): Normalized Energy Consumption (NEC) measures network efficiency by calculating the total energy consumed by all nodes relative to the number of successfully received data packets. Since transmission and reception demand significant energy, NEC reflects the energy cost per delivered packet. Lower NEC indicates better energy efficiency and prolonged network lifespan.

Network Lifetime: In MANETs, network lifetime is constrained by the finite energy of nodes, making efficient energy management crucial. When a node with low energy receives a route request (RREQ), it may drop the packet, leading to higher delays and reduced packet delivery. Network lifetime is defined as the duration until nodes deplete their energy and can no longer function.

* How does the EASP-AODV algorithm reduce link breakage in MANETs?

Response from the Author: The EASP-AODV algorithm reduces link breakage in MANETs by prioritizing nodes with higher residual energy, ensuring stable routes. It incorporates energy-aware path selection and sets energy thresholds to avoid nodes with low energy, which reduces the likelihood of link failures. The algorithm dynamically adjusts routes based on energy levels and node stability, preventing sudden link breakages. In case of failure, EASP-AODV employs localized route recovery to quickly find alternative paths. These measures help maintain stable communication and minimize link disruptions.

* What are the key differences between EASP-AODV and traditional AODV routing protocols?

Response from the Author: EASP-AODV improves upon AODV by incorporating residual energy into route selection, ensuring greater stability in MANETs. While AODV chooses the shortest path based on hop count, it overlooks node energy, leading to potential link failures. EASP-AODV prioritizes energy-rich nodes, reducing disruptions and enhancing network reliability. EASP-AODV improves upon AODV by integrating residual energy into route selection, ensuring stable and energy-efficient paths in MANETs. Unlike AODV, which selects routes based solely on hop count, EASP-AODV excludes low-energy nodes, proactively maintains routes, and repairs links using energy-aware alternatives. This approach reduces link failures, extends network lifetime, and enhances reliability.

* What simulation tool was used for evaluating the proposed algorithm?

Response from the Author: The EASP-AODV algorithm was evaluated using NS2, a widely used discrete-event simulator for modeling network protocols, including MANETs. NS2 enabled the simulation of network topologies and performance analysis based on metrics like packet delivery ratio, end-to-end delay, and normalized energy consumption. The results compared EASP-AODV with traditional AODV, highlighting its efficiency and stability.

* How does the EASP-AODV algorithm contribute to prolonging network lifetime?

Response from the Author: EASP-AODV extends MANET lifetime by prioritizing energy-efficient routes, avoiding low-energy nodes, and proactively maintaining stability. It reduces link failures, minimizes control overhead, and balances energy consumption across the network. These strategies enhance reliability and ensure prolonged network performance.

* How does the EASP-AODV algorithm compare to other existing energy-aware routing protocols in MANETs?

Response from the Author: EASP-AODV enhances energy-aware routing by integrating stable path selection, proactive maintenance, and balanced energy distribution. By preventing link failures and minimizing control overhead, it extends network lifetime more efficiently than existing protocols. This makes it ideal for energy-constrained MANET environments.

Reviewer #2:

The present work is interesting, however, some further edits are required to enhance the readability of the work.

1. Kindly make more work in the abstract to make it more attractive and according to the work. Response from the Author: We have made more attractive the abstract section according to the work.

2. The literature in the introduction is very less, I can recommend literature in a broad sense to improve it further and make connection of literature with the existing work, doi: 10.1109/TVT.2018.2828651, doi: 10.1109/TSC.2024.3478730, doi: 10.1109/MNET.2024.3350025, doi: 10.1109/TVT.2023.3304707, doi: 10.1109/TMM.2024.3394681, doi: https://doi.org/10.1287/moor.2022.1310, doi: https://doi.org/10.1145/3664655, doi: 10.1109/TCOMM.2024.3409539, doi: 10.1109/TMC.2024.3455417 and some other related work in the field should be searched.

Response from the Author: Response from the Author: We have added and reviewed the references in the literature section

3. Equation (1-4) should be properly referenced.

Response from the Author: We have properly referenced the equations

4. Before results and discussion some proper details of the method and related software used should be mentioned.

Response from the Author: The EASP-AODV algorithm enhances MANET longevity by integrating energy awareness into AODV, selecting energy-efficient routes, and dynamically adjusting paths based on real-time energy consumption. Simulated in NS-2, it demonstrated superior performance in packet delivery, energy efficiency, and network stability.

5. I recommend the authors improve the connection between sections and the obtained results should be properly linked with the method used.

Response from the Author: We have improved the connection between sections properly

6. Discussion should be enhanced in more aspects.

Response from the Author: This study proposed the EASP-AODV algorithm to enhance MANET lifetime by integrating energy awareness into route discovery and maintenance. Simulated in NS-2, EASP-AODV outperformed AODV in packet delivery, energy efficiency, and network stability by prioritizing energy-rich nodes and balancing load distribution. Future work will focus on optimizing scalability and refining energy models for real-world applicability.

Comments to the Author

1. If the authors have adequately addressed your comments raised in a previous round of review and you feel that this manuscript is now acceptable for publication, you may indicate that here to bypass the “Comments to the Author” section, enter your conflict of interest statement in the “Confidential to Editor” section, and submit your "Accept" recommendation.

Reviewer #1: All comments have been addressed

Reviewer #2: All comments have been addressed

2. Is the manuscript technically sound, and do the data support the conclusions?

Reviewer #1: Yes

Reviewer #2: Yes

3. Has the statistical analysis been performed appropriately and rigorously?

Reviewer #1: Yes

Reviewer #2: Yes

4. Have the authors made all data underlying the findings in their manuscript fully available?

Reviewer #1: Yes

Reviewer #2: Yes

5. Is the manuscript presented in an intelligible fashion and written in standard

---

## [Editor Report · Decision Letter 2]

27 Feb 2025

Energy Aware Stable Path Ad Hoc On-Demand Distance Vector Algorithm for Extending Network Lifetime of Mobile Ad hoc Networks

PONE-D-24-45310R2

Dear Dr. Girmaw,

We’re pleased to inform you that your manuscript has been judged scientifically suitable for publication and will be formally accepted for publication once it meets all outstanding technical requirements.

Kind regards,

Tawfik Al-Hadhrami, PhD

Academic Editor

PLOS ONE

Additional Editor Comments (optional):

The author has responded to all comments provided by the reviewers and I happy happy to accept the paper for publication.
---

## [Editor Report · Acceptance letter]

PONE-D-24-45310R2

PLOS ONE

Dear Dr. Girmaw,

I'm pleased to inform you that your manuscript has been deemed suitable for publication in PLOS ONE. Congratulations! Your manuscript is now being handed over to our production team.

Kind regards,

on behalf of

Dr. Tawfik Al-Hadhrami

Academic Editor

PLOS ONE